# LAD: Language Augmented Diffusion for Reinforcement Learning

**Edwin Zhang**[1], **Yujie Lu**[1], **William Wang**[1], and **Amy Zhang**[2]

[1]University of California, Santa Barbara.
{ete, yujielu, william}@cs.ucsb.edu
[2]Meta AI
amyzhang@meta.com

## Abstract

Learning skills from language provides a powerful avenue for generalization in re-inforcement learning, although it remains a challenging task as it requires agents to capture the complex interdependencies between language, actions, and states. In this paper, we propose leveraging **L**anguage **A**ugmented **D**iffusion models as a planner conditioned on language (LAD). We demonstrate the comparable performance of LAD with the state-of-the-art on the CALVIN language robotics benchmark with a much simpler architecture that contains no inductive biases specialized to robotics, achieving an average success rate (SR) of 72% compared to the best performance of 76%. We also conduct an analysis on the properties of language conditioned diffusion in reinforcement learning.

## 1 Introduction

It has been a longstanding dream of the AI community to be able to create a household robot that can follow natural language instructions and execute behaviors such as cleaning dishes or organizing the living room [1, 2, 3, 4, 5, 6, 7, 8, 9]. Incorporating language into Reinforcement Learning (RL) has great potential for generalization, enabling agents to utilize common sense priors across tasks and environments. Language provides an expressive abstraction of the environment and systematic generalization to new actions[10, 11, 12, 13, 14]. Given the recent progress in Natural Language Processing, how can one incorporate the powerful capabilities of language models and utilize them for downstream decision-making?

Language-conditioned policies (LCPs) are one class of policies in RL [15, 16, 13] that can be used to formulate this task, through the conditioning of behavior on natural language instruction. In this paper, we consider a novel approach for constructing LCPs by viewing the image as a sequence of state-actions rather than pixels, reformulating language conditioned decision-making into text-to-image generation.

Diffusion models such as DALL-E 2 [17] and GLIDE [18] have recently shown promise as generative models, with state-of-the-art text-to-image generation results demonstrating a surprisingly deep understanding of semantic relationships and generation of novel scenes. A key driver of the recent success in generative modeling is the usage of classifier-free guidance, which is amenable to the RL framework through the usage of language as a reward function. Inspired by the recent success of diffusion models in generative modeling [17, 18], we propose a new algorithm for LCPs (LAD) via latent diffusion models [19]. We demonstrate comparable performance to the state of the art on the CALVIN benchmark [20], with average success rate (SR) of 72% compared to the best performance of 76%.

Work done during a summer internship at Meta AI 2022. Language and Reinforcement Learning Workshop, 36th Conference on Neural Information Processing Systems (NeurIPS 2022).

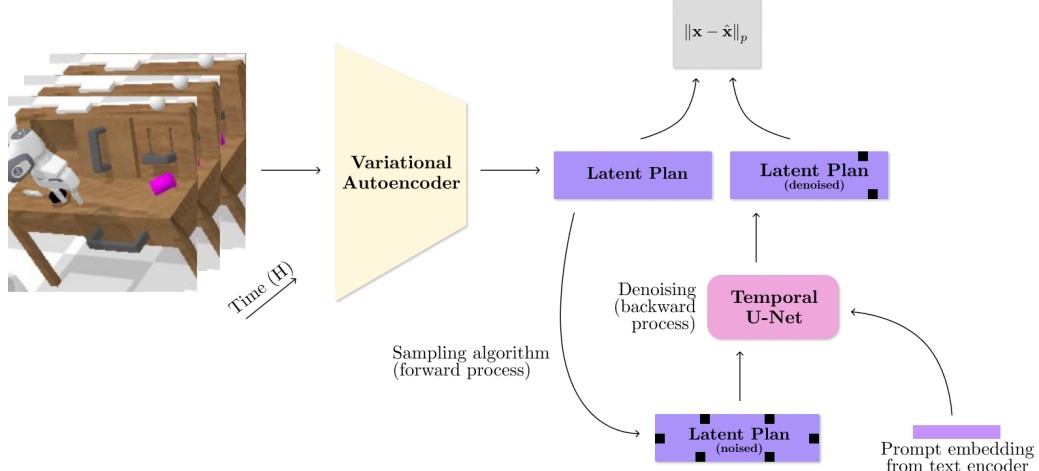

Figure 1: An overview of our training pipeline. The VAE is used to encode a sampled horizon of RGB observations into a lower dimensional latent space. We concatenate this sequence of states with the corresponding sequence of actions to construct the latent plan. We then noise the plan according to a uniformly sampled timestep from the diffusion process' variance schedule. We train a Temporal U-Net to reverse this process when conditioned on an encoded natural language instruction from an upstream language model, effectively learning how to conditionally denoise the latent plan. To train the U-Net, one can simply use the p-norm between the predicted latent plan and the ground truth latent plan as the loss. We set $p = 1$ in practice.

## 2 Language Augmented Diffusion

### 2.1 Method Overview

We consider a language-conditioned RL setting where we assume that the true reward function $\mathcal{R}$ is unknown, and must be inferred from a natural language instruction $L \in \mathscr{L}$. Formally, let $\mathcal{F}$ be the function space of $\mathcal{R}$. Then the goal becomes learning a $\phi : \mathscr{L} \mapsto \mathcal{F}$, and maximizing the policy objective conditioned on the reward function $\phi(L)$, $J(\pi(\cdot \mid s, \mathcal{R})) = \mathbb{E}_{a \sim \pi, s \sim p} \sum_{t=0}^{\infty} \gamma^t r_t$. Note that the space of tasks that can be specified by language is much larger than that of reward, due to the Markov restriction of the latter. For example, "pour the milk" and "pour the milk after five o'clock" are both valid instructions, but are indistinguishable from a reward function if the state does not contain temporal information. We assume access to a prior collected dataset $\mathcal{D}$ of $N$ annotated trajectories $\tau_i = \langle (s_0, a_0, ...s_T), L_i \rangle$. The language conditioned policy $\pi_\beta$, or the behavior policy. is defined to be the policy that generates the aforementioned dataset. In this paper, we assume access to a dataset of expert trajectories, such that $\pi_\beta$ = optimal policy $\pi^\star$. In this case, the policy objective reduces to imitation learning, or

$$\min_\pi \mathbb{E}_{s, \mathcal{R} \sim \mathcal{D}} \left[ D_{\text{KL}} \left( \pi_\beta(\cdot \mid s, \mathcal{R}), \pi(\cdot \mid s, \mathcal{R}) \right) \right]. \tag{1}$$

As we tackle the problem from a planning perspective, we define a trajectory generator as $\mathcal{P}$ and switch the atomic object from actions to trajectories $\tau$. Thus we aim to

$$\min_\mathcal{P} D_{\text{KL}} \left( \mathcal{P}_\beta(\tau \mid \mathcal{R}), \mathcal{P}(\tau \mid \mathcal{R}) \right)$$
$$= \min_\mathcal{P} \mathbb{E}_{\tau, \mathcal{R} \sim \mathcal{D}} \left[ \log \mathcal{P}_\beta(\tau \mid \mathcal{R}) - \log \mathcal{P}(\tau \mid \mathcal{R}) \right] \tag{2}$$

To model this, we turn to diffusion models [21]. Inspired by non-equilibrium thermodynamics, the common forms of diffusion models [22, 23, 24] propose modeling the data distribution $p(\tau)$ as a random process that steadily adds increasingly varied amounts of Gaussian noise to samples from $p(\tau)$ until the distribution converges to the standard normal. We denote the forward process as

$f(\boldsymbol{\tau}_t|\boldsymbol{\tau}_{t-1})$, with a sequence of variances $(\beta_0, \beta_1...\beta_T)$. We define $\alpha_t := 1-\beta_t$ and $\bar{\alpha}_t := \prod_{s=1}^{t} \alpha_s$.

$$f(\boldsymbol{\tau}_{1:T}|\boldsymbol{\tau}_0) = \prod_{t=1}^{T} f(\boldsymbol{\tau}_t|\boldsymbol{\tau}_{t-1}), \quad \text{where } f(\boldsymbol{\tau}_t|\boldsymbol{\tau}_{t-1}) = \mathcal{N}(\boldsymbol{\tau}_t; \sqrt{1-\beta_t}\boldsymbol{\tau}_{t-1}, \beta_t\mathbf{I}). \quad (3)$$

One can tractably reverse this process when conditioned on $\tau_0$, which allows for the construction of a sum of the typical variational lower bounds for learning the backward process' density function [22]. Since the backwards density also follows a Gaussian, it suffices to predict $\mu_\theta$ and $\Sigma_\theta$ which parameterize the backwards distribution:

$$p_\theta(\boldsymbol{\tau}_{t-1} \mid \boldsymbol{\tau}_t) = \mathcal{N}(\boldsymbol{\tau}_{t-1}; \boldsymbol{\mu}_\theta(\boldsymbol{\tau}_t, t), \boldsymbol{\Sigma}_\theta(\boldsymbol{\tau}_t, t)). \quad (4)$$

In practice, $\Sigma_\theta$ is often fixed to constants, but can also be learned through reparameterization. Following [23] we consider learning only $\mu_\theta$, which can be computed just as a function of $\tau_t$ and $\epsilon_\theta(\tau_t, t)$. One can derive that $\boldsymbol{\tau}_t = \sqrt{\bar{\alpha}_t}\boldsymbol{\tau}_0 + \sqrt{1-\bar{\alpha}_t}\epsilon$ for $\epsilon \sim \mathcal{N}(\mathbf{0}, \mathbf{I})$, through a successive reparameterization of (3) until arriving at $f(\boldsymbol{\tau}_t|\boldsymbol{\tau}_0)$. To sample from $p(\tau)$, we need only to learn $\epsilon_\theta$, which is done by regressing to the ground truth $\epsilon$ given by the tractable backwards density. Assuming we have $\epsilon_\theta$, we can then follow a Markov chain of updates that eventually converges to the original data distribution, in a procedure reminiscent of Stochastic Gradient Langevin Dynamics [25]:

$$\boldsymbol{\tau}_{t-1} = \frac{1}{\sqrt{1-\beta_t}} \left( \boldsymbol{\tau}_t - \frac{\beta_t}{\sqrt{1-\bar{\alpha}_t}}\boldsymbol{\epsilon}_\theta(\boldsymbol{\tau}_t, t) \right) + \sigma_t\mathbf{z}, \quad \text{where } \mathbf{z} \sim \mathcal{N}(\mathbf{0}, \mathbf{I}). \quad (5)$$

Thus, by using a variant of $\epsilon_\theta$ conditioned on language to denoise our latent plans, we can effectively model $-\nabla_\tau \mathcal{P}_\beta(\tau \mid \mathcal{R})$ with our diffusion model, iteratively guiding our generated trajectory towards the optimal trajectories conditioned on language.

## 2.2 Model Architecture

It is computationally infeasible to operate directly on the pixel space. Instead, we do planning in latent state space by first compressing the visual input with a $\beta$-TCVAE [26], similar to latent diffusion models [19]. We choose $\beta$-TCVAE for its efficiency in compression. The disentangled representation is a property that is also worth taking into consideration, and its effects on the denoising training process. It is still unclear whether disentanglement is beneficial or not. We adopt CLIP [27] as our textual encoder. CLIP is trained on large-scale image-text pairs and is able to align visual and textual input in its embedding space. Specifically, we use the Transformer variant [28] as the text encoder with the architecture modifications described in [29]. We use a modified temporal U-Net[30], which performs 1D convolution only across the time dimension, rather than the 2D convolution typical in text-to-image generation. This is motivated by our wish to preserve equivariance along the time dimension but not the state-action dimension. We modify the architecture in [30] by adding conditioning via cross attention in a fashion that resembles the latent diffusion model, but also uses a temporal convolution for projection to the token embeddings rather than the traditional 2D convolution. We use DDIM [24] during inference for increased computational efficiency and faster planning. DDIM uses strided sampling and is able to capture almost the same level of fidelity as typical DDPM sampling [23] with an order of magnitude speedup. For rolling out the latent plans generated by the denoiser, we resample a new plan with the frequency of $H$, until either the task is completed successfully or the maximum timestep is reached. In between samples, we roll out the open-loop plan without replanning. We set the total time horizon equal to $3H$ in our experiments, which means at maximum we will sample three latent plans from the model.

## 3 Experiments

### 3.1 Dataset and Metric

We evaluate a subset of the CALVIN benchmark [20] due to time, selecting the five most common subtasks (i.e tasks with the most pre-collected data) that do not belong to the same subtask category. This metric allows for fast experimentation since there is significantly less data to converge on, while still covering a diverse and broad range of tasks as we enforce that each task will belong to a different category. After pretraining our VAE on all data, we freeze the autoencoder and train the

Figure 2: An overview of our inference pipeline. We pass sampled noise to our denoising autoencoder along with an encoded language prompt. The autoencoder is instantiated as a temporal U-Net. By repeating this process iteratively, we are able to generate high-fidelity latent plans conditioned on language. We are also able to decode the latent states into pixel space to analyze and interpret the plans generated by the denoising autoencoder.

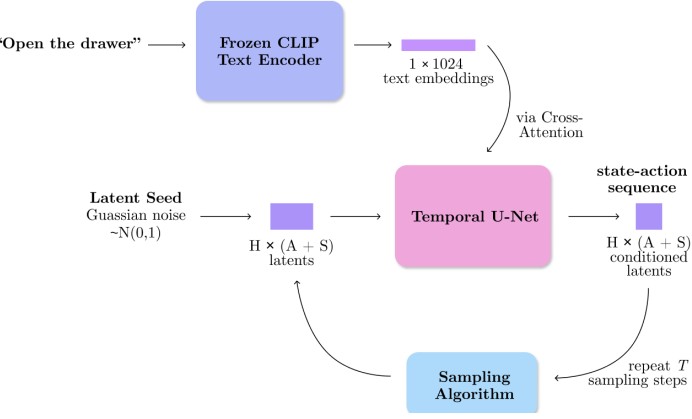

U-Net just with data from the five tasks. This setup is motivated by the assumption that we only wish to evaluate the performance of the downstream decision-making, so we assume that there already exists a robust pre-trained autoencoder. In order to provide a fair comparison, we obtain results from our comparisons by freezing the autoencoders of their final checkpoint which was trained on all data and training the rest of the model on just the five tasks. We roll out all evaluated policies for 5 trajectories per task, for a total of 25 rollouts per policy. We compare the two strongest models released on this benchmark so far, HULC and MCIL [9, 31] All comparisons are trained in their official repository[1].

| Task | MCIL | HULC | LAD |
|---|---|---|---|
| Place in Slider | 1.0 | 0.8 | 1.0 |
| Open Drawer | 1.0 | 1.0 | 1.0 |
| Move Slider Right | 0.4 | 1.0 | 1.0 |
| Stack Block | 0.2 | 0.4 | 0.2 |
| Lift Blue Block Table | 0.4 | 0.6 | 0.4 |
| **Total (avg)** | **0.60** | **0.76** | **0.72** |

Table 1: Comparison of success rates between our diffusion model and prior benchmarks. Although our absolute performance does not beat the prior SOTA, we note that HULC is a significantly more complex model containing many inductive biases for robotics that may not be transferable to other RL environments, such as using a separate logistic loss for modeling the gripper action.

## 3.2 Effects of Diffusion

An interesting phenomenon we observe when rolling out with the diffusion model is its ability to robustly model arbitrary starting points of trajectories, leading to an ability to recover from failures in rollouts and attempt tasks again. For this reason, we find that simply by replanning more times instead of just rolling out a single plan improves the performance of LAD. There is much more to explore in this direction, and future rollout strategies that include conditioning on more than the last state or replanning more often than just once every H timesteps will likely lead to substantial gains in performance.

## 4 Conclusion

Learning the atomic sub-skills is critical to solving the multi-task planning problem and enabling the solving of more complex and open environments through state and temporal abstraction. We explore reformulating the language-conditioned planning process as the text-to-plan generation to better learn the alignment between language and state-action pairs. Experiments and qualitative analysis demonstrate the simplicity and effectiveness of our model. Future work looks to extend to the long horizon setting and further probing of the generalization and compositional capabilities of the model through classifier-free guidance or improved sampling methods, as well as incorporating value functions for guiding towards optimality.

---

[1] https://github.com/lukashermann/hulc

**Acknowledgments**

We would like to thank Ashley Jiang for creating the main figures, Michael Janner and Weixi Feng for fruitful discussions, and Oier Mees and Luka Shermann for clarifications and support on the CALVIN benchmark.

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

# A    Qualitative Analysis

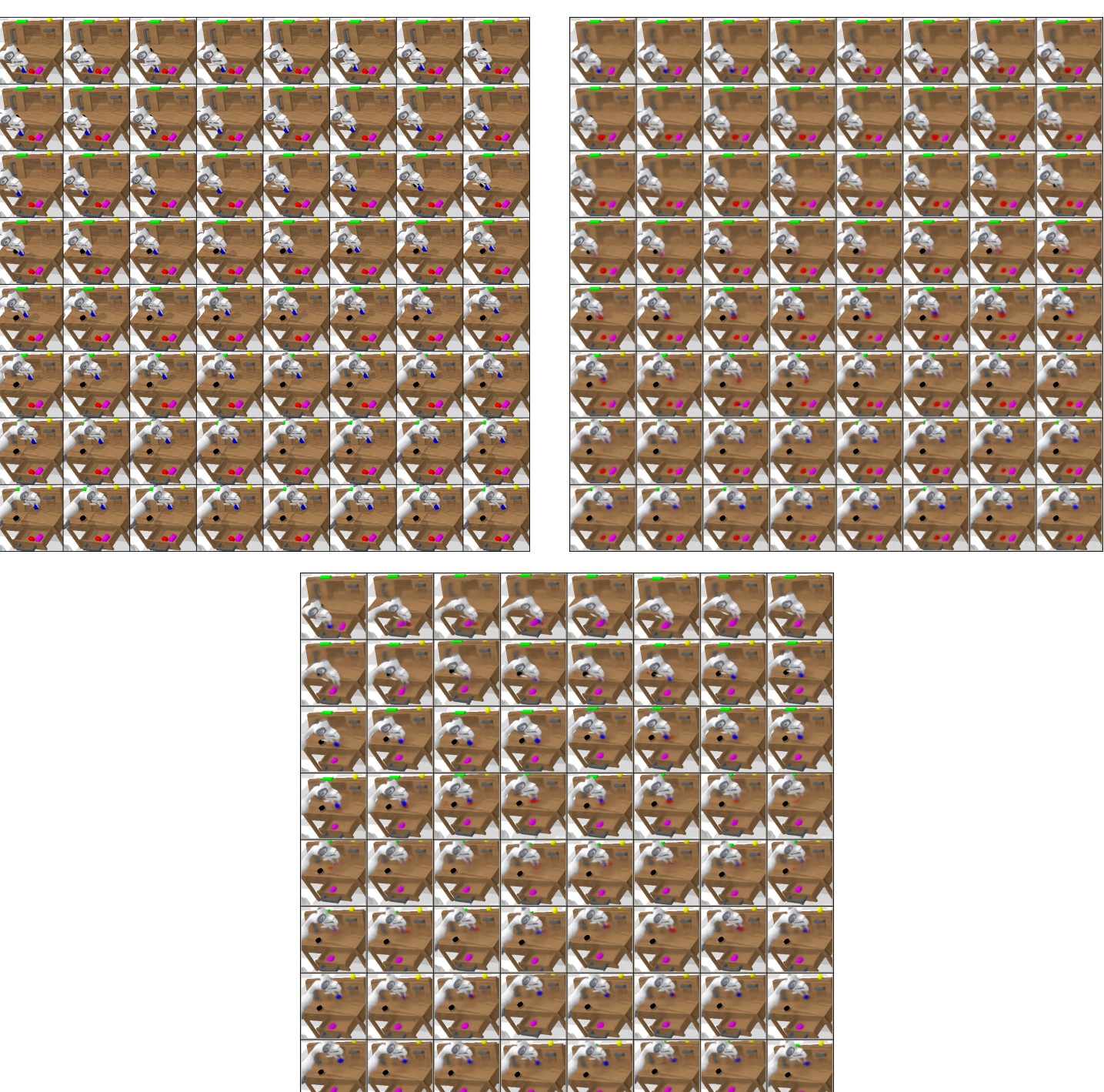

Figure 4: Comparison of a ground truth trajectory for the "place in slider" task (left), the reconstructed trajectory (right), and the generated trajectory through denoising (bottom).

# B    Background and Related Work

**Reinforcement Learning**

We formulate the RL framework as Markov decision process (MDP) $\mathcal{M} = (\mathcal{S}, \mathcal{A}, \mathcal{R}, \gamma, p)$, with state space $\mathcal{S}$, action space $\mathcal{A}$, reward function $\mathcal{R}$, discount factor $\gamma$, and transition dynamics $p$. At each time step $t$, agents observe a state $s \in \mathcal{S} \subseteq \mathbb{R}^n$, take an action $a \in \mathcal{A} \subseteq \mathbb{R}^m$, and transition to a new state $s'$ with reward $r$ following $s', r \sim p(\cdot, \cdot | s, a)$. The goal of RL is then to learn either a deterministic policy $\pi : \mathcal{S} \mapsto \mathcal{A}$ or a stochastic policy where $a \sim \pi(\cdot | s)$ with $a \in \mathcal{A}$ that maximises the policy objective, $J(\pi) = \mathbb{E}_{a \sim \pi, s' \sim p} \sum_{t=0}^{\infty} \gamma^t r_t$. Notice this is nothing but the expected discounted cumulative reward or expected return.

**Language-Conditioned Policy** Language-conditioned policies [15, 16, 13] have been explored in the reinforcement learning community to improve the ability to abstract the goal and the generalization to a new environment [10, 11, 12, 13]. However, these LCPs still struggle with long-horizon language commands. We are the first to leverage the advantage of the diffusion model in compositionality and long-horizon decision-making to address such challenges.

**Diffusion in Offline Reinforcement Learning**

Given the success of the denoising diffusion probabilistic models [23] (DPM) applied in text-to-image synthesis [32], the DPM has been further explored in both discrete and continuous data domains, including image and video synthesis [33, 34], text generation [35], and time series [36]. Diffusion planning [30] first proposed to transform the planning problem into inpainting and utilize diffusion models to solve the problem. Specifically, they diffuse the state and actions jointly to implement imitation learning and goal-conditioned reinforcement learning. This leverage the diffusion to solve long-horizon and compositionality issue in planning. Instead of predicting the whole trajectory for each state, [37] apply the diffusion model to sample a single action at a time conditioned with states. Analogous to the Diffuser [30] in trajectory-planning perspective and [37] in offline policy-optimization perspective, we are in the language-grounding perspective.

