# OpenReview forum: "LAD: Language Augmented Diffusion for Reinforcement Learning"
_NeurIPS.cc/2022/Workshop/LaReL — LaReL 2022_

### Official Review · Reviewer_ujTo · 2022-10-05
**I don't understand at all what the authors are trying to do**

**Rating:** 5
**Confidence:** 1

**Review:**

I cannot provide a summary of the paper as, to me, the authors did not explain at all what they are trying to do.

They have instruction-following agents, they want to build latent plans (to solve tasks in the CALVIN robotics benchmark, which is not described at all), and they "consider a novel approach for constructing LCPs by viewing the image as a sequence of
state-actions rather than pixels, [they] reformulate language conditioned decision-making into text-to-image generation."
I don't know at all where this image comes from and what is tis role in the story.

A guess might be that from the language instruction they generate images, then these images are used in the latent plan? It is neither explained anywhere what they do nor their general approach to do it.

Instead, the "method overview" is a technical description using unexplained equations where part of the symbols are not defined.

From the caption of Fig 1, we discover that there is a "ground truth latent plan". How do the authors get this?

More local points:

"Note that the space of tasks that can be specified by language is much larger than that of reward, due to the Markov restriction of the latter. For example, “pour the milk” and “pour the milk after five o’clock” are both valid instructions, but are indistinguishable from a reward function if the state does not contain temporal information."
-> actually, if the agent has no access to temporal information, the language instruction "after five o'clock" is useless too.

All equations which finish a sentence should be followed by a dot.

The language conditioned policy π β , or the behavior policy. is -> remove dot

In (2), P_\beta is not explained. And the intuition behind (2) should be described with a sentence.

So, in short, for a 4 pages paper, the authors should rather try to explain their work for a naive reader rather than sticking to an unexplained technical description.

---

### Official Review · Reviewer_7Kwj · 2022-10-12

**Rating:** 7
**Confidence:** 3

**Review:**

This paper presents an approach to use language-conditioned diffusion to learn a generative model for the latent representations of trajectories, which are then decoded into trajectories to produce roll-outs.

I think this is an interesting attempt to see the potential of diffusion-style models in the RL setting.  A few experiments in the CALVIN benchmark show some early signs of promise for this approach.

Writing-wise I think the paper could use a bit more space to describe the model details to improve clarity rather than going through the formulation of the diffusion process which is at this point relatively standard.

Going forward, adding in reward signals to do RL on top of the diffusion process could be interesting to explore.  Alternatively, using the learned generative model as the reward function to train a policy could also be interesting, rather than using the generative model plus the decoder as the policy directly.

---

### Decision · Program_Chairs · 2022-10-20

Accept